# Dynamic multiphase semi-crystalline polymers based on thermally reversible pyrazole-urea bonds

Wen-Xing Liu [1,3], Zhusheng Yang[1], Zhi Qiao[1], Long Zhang[2]*, Ning Zhao[1]*, Sanzhong Luo[2] & Jian Xu [1]*

Constructing responsive and adaptive materials by dynamic covalent bonds is an attractive strategy in material design. Here, we present a kind of dynamic covalent polyureas which can be prepared from the highly efficient polyaddition reaction of pyrazoles and diisocyanates at ambient temperature in the absence of a catalyst. Owing to multiphase structural design, poly (pyrazole-ureas) (PPzUs) show excellent mechanical properties and unique crystallization behavior. Besides, the crosslinked PPzUs can be successfully recycled upon heating (~130 °C) and the molecular-level blending of polyurea and polyurethane is realized. Theoretical studies prove that the reversibility of pyrazole-urea bonds (PzUBs) arises from the unique aromatic nature of pyrazole and the N-assisting intramolecular hydrogen transfer process. The PzUBs could further broaden the scope of dynamic covalent bonds and are very promising in the fields of dynamic materials.

---

[1] Beijing National Laboratory for Molecular Sciences (BNLMS), Laboratory of Polymer Physics and Chemistry, Institute of Chemistry, Chinese Academy of Sciences, Beijing 100190, China. [2] Center of Basic Molecular Science (CBMS), Department of Chemistry, Tsinghua University, Beijing 100084, China. [3]Present address: Department of Organic Chemistry, University of Geneva, Geneva CH-1211, Switzerland. *email: zhanglong@iccas.ac.cn; zhaoning@iccas.ac. cn; jxu@iccas.ac.cn

D ynamic polymer materials, which rely on the covalent bonds capable of undergoing the reversible formation and cleavage under appropriate conditions, are able to reorganize their macromolecular architectures or compositions, and thus produce a macroscopic response to the environment and surrounding stimuli[1–3]. In recent years, great efforts have been made to create complex, smart, and adaptive polymeric materials that allow access to topological/constitutional structure transformation[4], controlled assembly[5], shape memory[6,7], self-healing, and recyclability[8–33], to name a few. Although in this context, the library of dynamic polymers has extended rapidly, it remains a challenge to design mechanically robust dynamic polymers with high strength, extensionality, and malleability.

Isocyanate-based polymers, polyureas, or polyurethanes, are widely used in coatings, elastomers, adhesives, foams, and fibers[34]. Recently, by introducing oxime moiety into urethane bonds, we developed the dynamic poly(oxime-urethanes) (POUs)[35], which have shown excellent self-healing ability, malleability, and mechanical properties[31,35]. Compared with urethane bonds, urea bonds are more stable and irreversible due to the dual resonance stabilized structure. Typically, the neutral hydrolysis half-time of polyureas is >$10^4$ years[36]. Harsh conditions are required to cleave urea bonds, such as acidic or basic solutions, high temperature, and catalysts[37–39], which limits their applications in dynamic covalent materials. Metal catalyst such as zinc acetate shows effectiveness to increase the reversibility of normal urea bonds[40]. In addition, by introducing bulky substituents to a urea nitrogen atom, the hindered urea bonds are significantly destabilized owing to the steric inhibition of resonance via hindering the orbital co-planarity of the urea bond, thus resulting in reversibly dissociating to isocyanate and amine under ambient condition without a catalyst[41–43]. However, due to the high sensitivity of isocyanate intermediate to moisture, this system may suffer from high susceptibility to hydrolysis[43].

As stable aromatic heterocyclic compounds, pyrazoles have been used as building blocks in some material fields[44,45]. Owing to the aromaticity of pyrazoles, opposed resonance occurs in pyrazole-amide bonds and the conjugation effect between the pyrazole nitrogen and the carbonyl group is slightly diminished (Fig. 1a)[46]. This interesting N-acylpyrazole chemistry inspires us that pyrazole may be a potential candidate for the design of dynamic amide-containing polymers.

Here, we present the reversible pyrazole–urea bonds (PzUBs), which are applied for the design of a series of mechanically robust dynamic polymers, poly(pyrazole–ureas) (PPzUs). PPzUs are stable at room temperature, and spontaneously reversible at elevated temperatures. Unlike previously reported dynamic polymers, whose crystallization is inhibited because of structure design with bulky monomers or high-cross-linking density, the highly cross-linked PPzU thermosets are semi-crystalline polymers. The crystallization behavior of PPzUs offers substantial advantages in terms of strength, toughness, and solvent resistance in contrast to the amorphous dynamic polymers.

## Results

### Dynamic chemistry of pyrazole–urea bonds.
Although kinetic and thermodynamic behaviors are the underlying features of dynamic reactions, these chemical characteristics of PzUBs have not been studied up to now[47,48]. We first investigated the addition reaction between octyl isocyanate **1b** and 4-methylpyrazole **1c** (Fig. 1b), whose rate was monitored through the change of NCO infrared absorption. The reaction could almost proceed to completion within 25 min at 30 °C in various aprotic solvents, with the second binding rate constants $k_b$ in the range of 4.2–9.7 × $10^{-2}$ $M^{-1} s^{-1}$ (rate = $k_b$[NCO][pyrazole],

Supplementary Figs. 1–2) and an Arrhenius activation energy $E_{a,b}$ of 11.3 kcal $mol^{-1}$ (Table 1; Supplementary Fig. 3). What is more, PzUBs show good stability at ambient temperature (Supplementary Fig. 4). However, a 6% thermal dissociation of **1bc** was obtained in several minutes by $^1$H NMR at 110 °C with a dissociation equilibrium constant $K_d$ of 1.3 × $10^{-3}$ M and a dissociation enthalpy $\Delta H_d$ of 18.5 kcal $mol^{-1}$ (Table 1; Supplementary Fig. 5).

Then a more detailed mechanism investigation was undertaken by using aryl substituted pyrazole–ureas and exploring thermodynamics of their thermal dissociation reactions. A Hammett analysis of the dissociation equilibriums revealed that the dissociation step is hindered by electron-withdrawing groups, such as Cl and $CF_3$ (Supplementary Fig. 6). A linear free energy relationship was found with a negative reaction slope ($\rho = -1.0$), indicating that a fast proton transfer may occur first in the reversed process and the C–N bond breaking step is rate-determining during which the negative charge on the urea moiety is dramatically reduced.

To quantify the kinetics of the dissociation reaction, the exchange reaction of **1ac** and **1bd** to produce **1ad** and **1bc** was followed by liquid chromatography at different temperatures (Fig. 1c; Supplementary Fig. 7). As expected, the equilibrium mixture containing equimolar amounts of all four pyrazole–ureas was obtained after 150 min at 120 °C or 2000 min at 90 °C. Besides, the dissociation rate constant $k_d$ of 0.12 $h^{-1}$ at 90 °C and the dissociation activation energy $E_{a,d}$ of 26.0 kcal $mol^{-1}$ were found (Table 1, Supplementary Figs. 8–9), which are close to those of other thermally reversible bonds[1,8,35]. Above studies show that PzUBs with fast formation process and good stability at room temperature can undergo evidently thermal reversibility without any catalyst.

### Theoretical studies.
Density functional theory (DFT) calculations were performed to gain further insight into the formation and dynamic nature of PzUBs with pyrazole and methyl isocyanate (MeNCO) as model reactants at the theory level of B3LYP/6-311 + + G(2df,2pd)//B3LYP/6-31 + G(d,p). Theoretical studies indicated that after the formation of a complex between pyrazole and isocyanate, nucleophilic attack of the pyridine-type nitrogen in pyrazole to MeNCO, via transition state **TS1** with an activation enthalpy of 12.8 kcal $mol^{-1}$ and an activation free energy of 24.4 kcal $mol^{-1}$, leads to intermediate **Int2** (3.7 kcal $mol^{-1}$ below **TS1**). The subsequent proton shift process smoothly produces pyrazole–urea **P1** with an activation enthalpy of 3.3 kcal $mol^{-1}$, which is exothermic by 23.0 kcal $mol^{-1}$. According to these calculations, this pathway with nucleophilic addition as rate-limiting step accords with the first-order kinetics for both pyrazole and isocyanate as well as the Hammett analysis (Supplementary Figs. 1 and 6), and the reverse reaction requires an activation enthalpy of 27.9 kcal $mol^{-1}$ (Fig. 2a). These activation parameters are close to those obtained by experiments (Supplementary Figs. 10–11, Supplementary Tables 1–2).

For further consideration, resonance energies ($E_R$) of the amide bond in different ureas were calculated to account for the distinctions in dynamicity at the level of B3LYP/6-31 + G(d) (Fig. 2b)[49]. Resonance energy in pyrazole–urea (5.6 kcal $mol^{-1}$) is lower than that in normal urea (7.2 kcal $mol^{-1}$), but is higher than that in hindered urea (3.9 kcal $mol^{-1}$), which is consistent with the relative stability of the corresponding ureas. Therefore, the compatibility of stability and reversibility of PzUBs originates from the aromatic character of pyrazole moderately weakening the resonance stabilization and the presence of adjacent nitrogen atom facilitating the intramolecular 1,4-hydrogen transfer process (Supplementary Figs. 12–14).

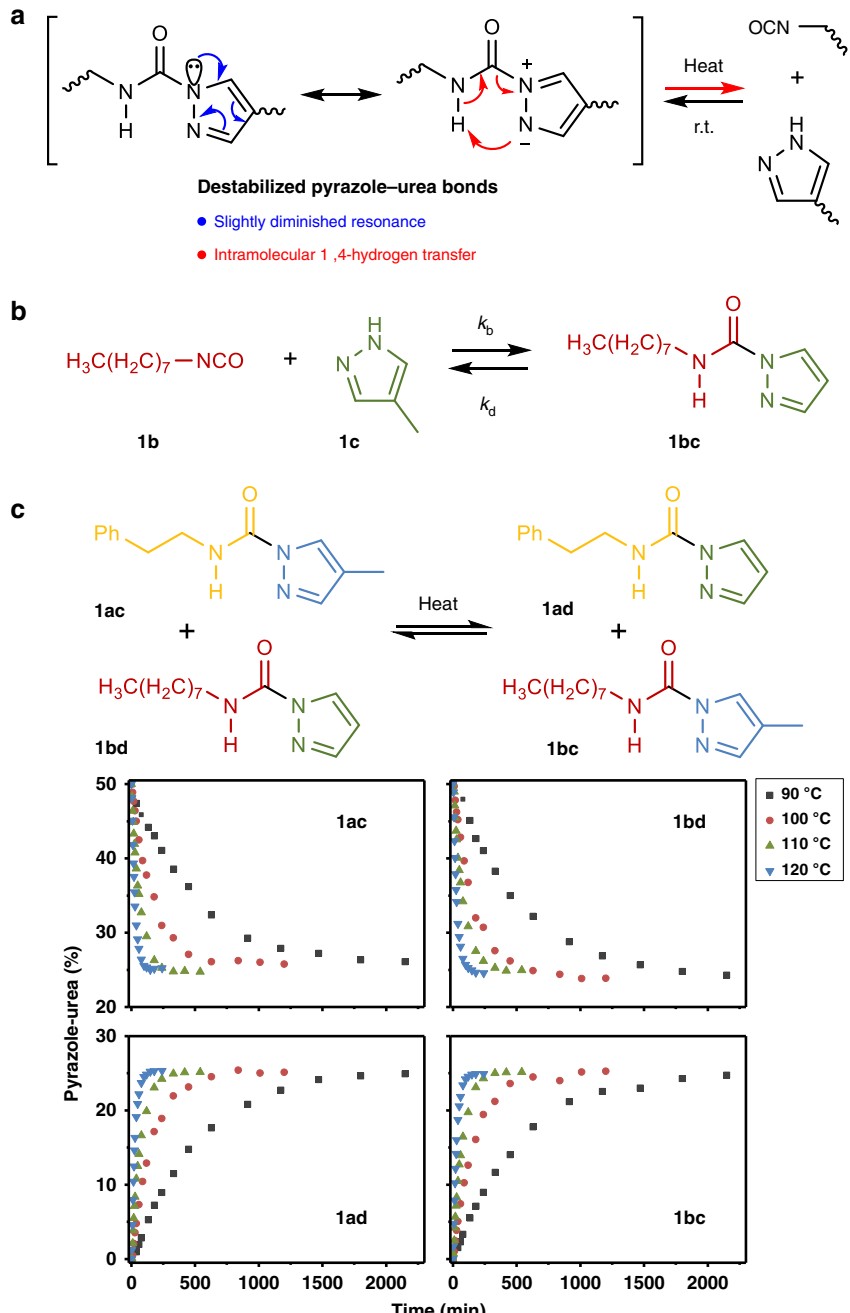

**Fig. 1** The dynamic nature of pyrazole–urea bonds. **a** Destabilized pyrazole–urea bonds through opposed resonance and kinetically favored intramolecular hydrogen transfer. **b** The association and dissociation of **1bc**. **c** The model exchange reaction of **1ac** and **1bd** to produce **1ad** and **1bc** as a function of time upon heating at different temperatures

**Table 1 Kinetic and thermodynamic parameters of PzUBs**

| $k_b$ (303 K, M$^{-1}$•s$^{-1}$)[a] | $E_{a,b}$ (kcal•mol$^{-1}$)[a] | $k_d$ (363 K, h$^{-1}$)[b] | $E_{a,d}$ (kcal•mol$^{-1}$)[b] | $K_d$ (383 K, M)[c] |
|---|---|---|---|---|
| $6.4 \times 10^{-2}$ | 11.3 | 0.12 | 26.0 | $1.3 \times 10^{-3}$ |

[a]The reaction was performed with **1b** (0.80 mmol) and **1c** (0.80 mmol) in CH$_2$Cl$_2$ (2 mL)
[b]Obtained from the model exchange reaction in bulk
[c]The dissociation of **1bc** (0.21 mmol) was carried in d$_6$-DMSO (0.5 mL)

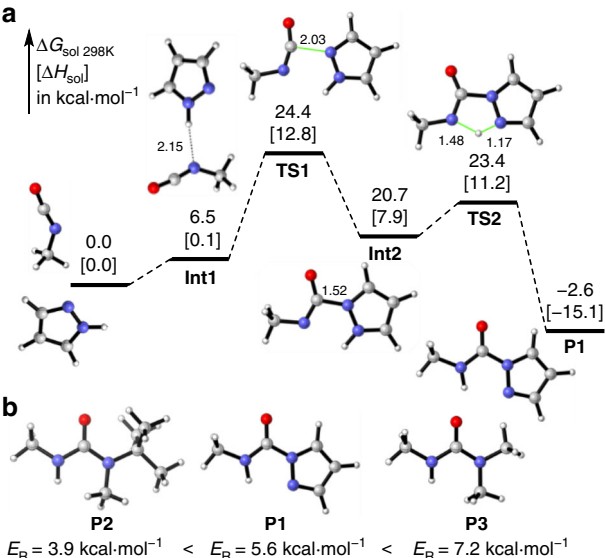

**Fig. 2** Theoretical investigations for the formation and dynamicity of PzUBs. **a** DFT calculated reaction profiles for the reaction of MeNCO and pyrazole (distances in Å). **b** Resonance energies of pyrazole–urea **P1**, hindered urea **P2** and normal urea **P3**

**Synthesis and characterization of poly(pyrazole–ureas).** With the chemistry of PzUBs in hand, we intended to fabricate dynamic covalent polymeric materials based on PzUBs. As microphase-separated polymer structures being composed of hard segments (HS) and soft segments (SS) could exhibit both good strength and high extensibility[50], multiphase PPzUs were designed using multifunctional pyrazoles (**4** and **5**) and hexamethylene diisocyanate (HDI) as monomers (Fig. 3). The flexible poly(thio)ether chain containing cross-linker would act as the SS, while the HDI and pyrazole constitute the HS. In this design, not only the symmetric structure of HDI and aromatic planarity of pyrazole would contribute to the regular packing of the molecular chain in HS but the installation of covalent cross-linking in SS also reduce the inhibitory effect on HS crystallization.

The monomers can be simply prepared from 4-pyrazolecarboxylic acid 2 within two steps through esterification reaction and thiol-ene reaction. Then, linear or cross-linked PPzUs (6 or 7a–c) with variable formulas were synthesized through the polymerization of multifunctional pyrazoles and HDI at room temperature in CHCl3 (Fig. 4a; Supplementary Figs. 15–16).

Differential scanning calorimetry (DSC) thermograms disclose that one obvious endothermic peak is detected for each of the four PPzUs (Fig. 4b; Supplementary Fig. 17), which can be assigned to the melting of partially crystallized HS from pyrazole–urea domains. Of particular note is that PPzU **7c** with the highest cross-linking degree (average molecular weight

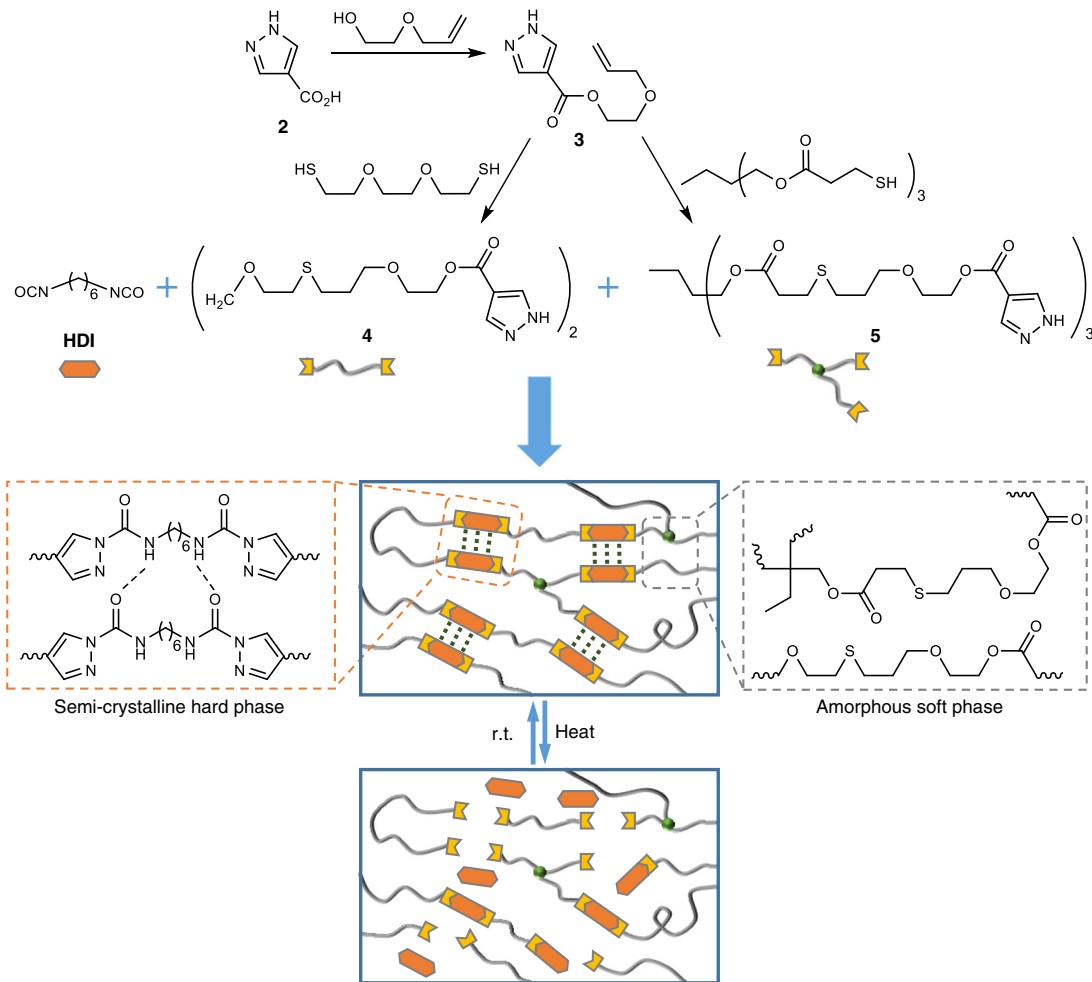

**Fig. 3** Design and synthesis of dynamic multiphase poly(pyrazole–ureas)

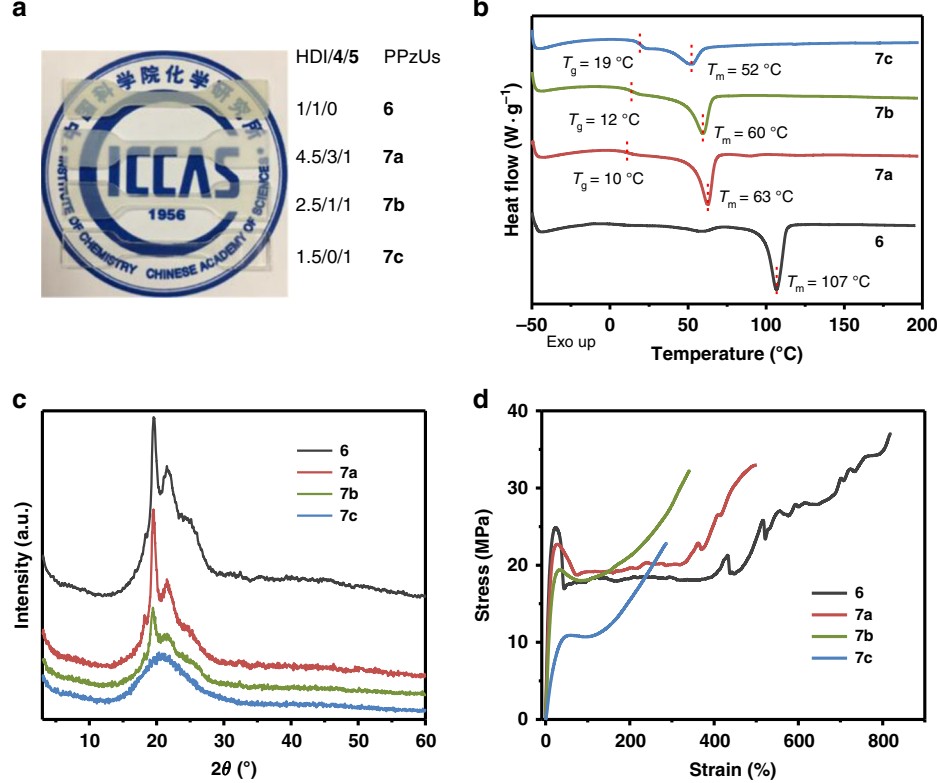

**Fig. 4** Characterization of semi-crystalline poly(pyrazole–ureas). **a** Photographs for PPzUs (**6** and **7a–c**) with different formulas. **b** DSC curves of the second heating runs for PPzUs. **c** XRD analysis of PPzUs. **d** Stress–strain curves of the as-prepared PPzUs

between cross-links ($M_c$) = 1.7 kg mol$^{-1}$, Supplementary Table 3) is still semi-crystalline although cross-linking process indeed interferes with the crystallization. This is markedly different from formerly reported ones which always exhibited amorphous behavior[8–33,40–43]. With the increase of cross-linking density (Supplementary Table 3), the decrease of the intensities of crystalline peaks ($2\theta = 19°$ and $21°$) in X-ray diffraction (XRD) patterns (Fig. 4c) and the gradual disappearance of birefringence (Supplementary Fig. 18) are both observed. Dynamic mechanical analysis reveals that PPzUs possess three main thermal transitions (Supplementary Fig. 19), where the first one corresponds to the glass transition ($T_{g,SS} \approx -10$ °C) of SS, while the last two should be the glass transition ($T_{g,HS} \approx 30$ °C) and melting of HS ($T_{m,HS} \approx 110$ °C for **6**, **7a−b**; ≈60 °C for **7c**). Small-angle X-ray scattering (SAXS) shows that except for **7c**, other PPzUs exhibit a broad primary scattering peak with the corresponding $d$-spacing values falling into the range of 6.4–8.8 nm (Supplementary Fig. 20). The lack of higher orders of scattering peaks suggests that the phase separation is likely short-range correlated, which is rational in view of the fact that these PPzUs are not typical block copolymers[51]. Furthermore, atomic force microscopy images also proved the microphase separation between soft and hard domains (Supplementary Fig. 21). Gratifyingly, the crystallization endows PPzUs with excellent solvent resistance (Supplementary Fig. 15) and mechanical properties. Representatively, PPzU **6** has a tensile strength ($\sigma_b$) of 35.9 ± 0.9 MPa and a strain-at-break ($\varepsilon_b$) of 805 ± 9% (Fig. 4d; Supplementary Table 4). In comparison with dynamic hindered polyureas[43], PPzUs show excellent hydrolytic stability at 37 °C within 6 days (Supplementary Fig. 22). These observations indicate that PPzUs are a kind of high-performance semi-crystalline polymers.

**Thermal reversibility of poly(pyrazole–ureas).** Next, we pay our attention to evaluate the dynamicity of PPzUs. To restrain the

crystallization and increase the solubility, PPzU **8** was prepared from 1,3-bis(isocyanatomethyl)-cyclohexane (mixture of cis/trans isomers) instead of HDI. By tuning the monomer ratio, PPzU **8** can be initially depolymerized at elevated temperature and then repolymerized at low temperature (Supplementary Fig. 23), which was also observed in our recently reported dynamic poly(oxime-urethanes) (POUs)[35]. Interestingly, after the mixture of PPzU **8** ($M_n = 12$ kDa) and POU **9** ($M_n = 58$ kDa, for synthesis see Supplementary Methods) was heated at 110 °C for 3 h in $N,N$-dimethylformamide (DMF), the original peaks derived from **8** and **9** disappeared and fused into a new unimodal peak of poly(urethane-urea) **10** ($M_n = 18$ kDa, Fig. 5a, b). The successful reorganization and molecular-level blending of polyurethane and polyurea through macromolecule interchange reaction are undoubtedly attributed to the common isocyanate intermediate in the two reversible reactions. Different from traditional polymer blending methods, this artful methodology using dynamic covalent bonds provides an approach to polymer blends via the interchange reaction of polymer chains.

The plasticity of the cross-linked PPzU **7c** was studied by temperature-dependent stress-relaxation analysis, and the results showed that higher temperature accelerates faster relaxation (Fig. 5c). The characteristic relaxation time $\tau^*$ (where $G/G_0 = e^{-1}$) follows the Arrhenius law, and an activation energy $E_{a,r}$ of 24.5 kcal mol$^{-1}$ was calculated (Supplementary Fig. 24), which is in good conformity with that obtained for the model compounds ($E_{a,t} = 26.4$ kcal mol$^{-1}$, Supplementary Fig. 7). The thermal reprocessability of the cross-linked PPzUs was characterized by uniaxial tensile testing on both pristine and recycled dogbone samples. Representatively, PPzU **7c** can almost fully recover its mechanical properties on $\sigma_b$ and $\varepsilon_b$ when being remolded at 130 °C for 30 min, even after three cycles (Fig. 5d; Supplementary Fig. 25 and Supplementary Table 4). Noteworthily, the obvious change on yield stress and Young's modulus of the original and the first recycled samples could be observed, especially

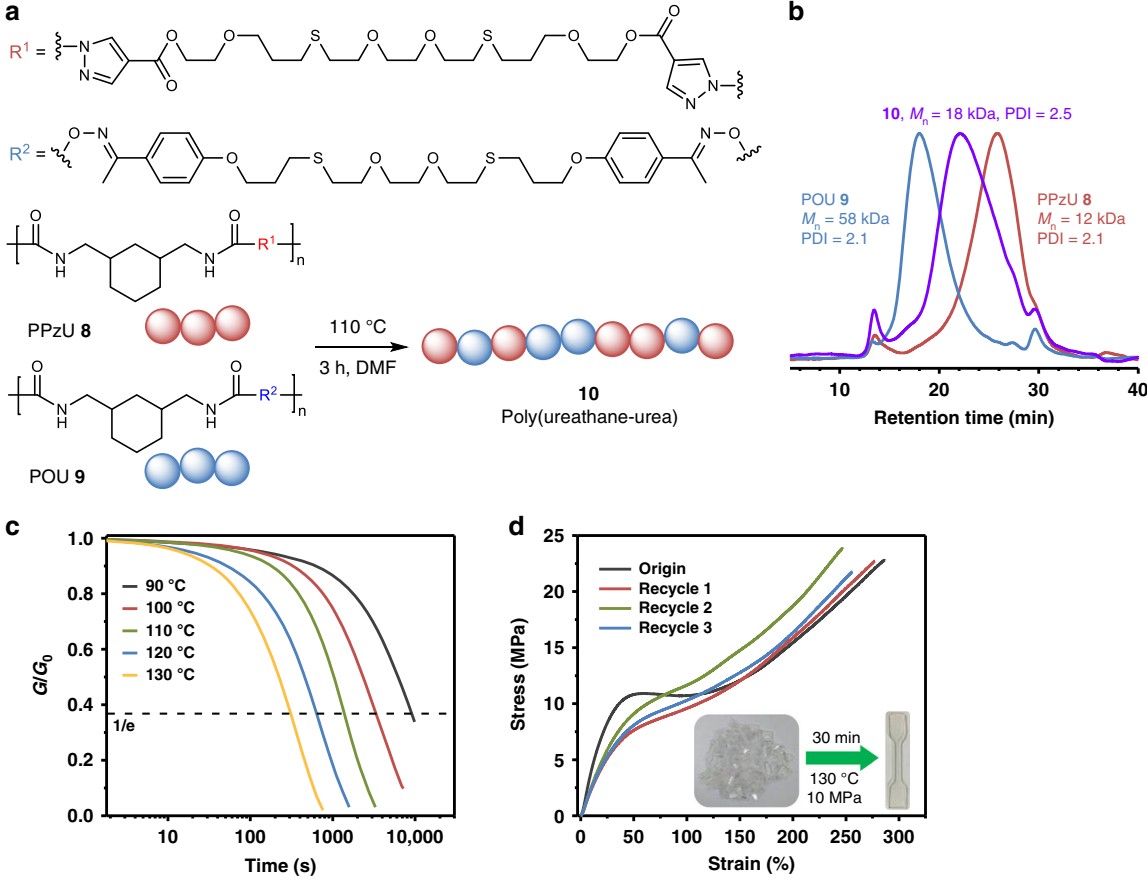

**Fig. 5** Thermal reversibility of poly(pyrazole–ureas). **a** Macromolecular interchange reaction of PPzU **8** and POU **9** monitored by (**b**) gel permeation chromatography. **c** Normalized stress-relaxation analysis of PPzU **7c**. **d** Recyclability of PPzU **7c** evaluated by tensile test (the inset shows compression molding of **7c**)

for PPzUs **7a** and **7b**. Both the complete recovery of plateau modulus and the absence of isocyanate IR absorption for all recycled PPzU thermosets exclude the possibility of the irreversible breaking of PzUBs (Supplementary Figs. 19 and 26). The slight differences in tensile curves of recycled samples imply that the change of mechanical properties is more likely caused by the distinction of crystallization behavior between the original preparation process and the thermal recycling experiment, which is also supported by the SAXS and XRD experiments (Supplementary Figs. 20 and 26).

## Discussion

In conclusion, we have successfully introduced the dynamic ability of pyrazole–urea bonds, which can undergo rapidly thermal dissociation or exchange reactions. Moreover, the dynamic pyrazole–urea bonds are utilized for the synthesis of a type of mechanically robust semi-crystalline dynamic polymers, poly (pyrazole–ureas). The formation of poly(pyrazole–ureas) between isocyanate and pyrazole is fast, simple, and catalyst-free at room temperature. The unique crystallization property in poly (pyrazole–urea) thermosets is rarely reported in most other highly cross-linked dynamic polymers. Considering the widely existed isocyanate-based systems, the pyrazole–urea bonds will offer a valuable platform for a range of applications, including materials design with dynamic adaptive capacity.

## Methods

**General information**. For synthetic procedures and NMR spectra of compounds and linear polymers, see Supplementary Methods and Supplementary Figs. 27–43. For details on kinetic and thermodynamic studies, see Supplementary Note 1. Computational methods can be found in Supplementary Methods and

Supplementary Note 2. Cartesian coordinates of all molecules are given in Supplementary Note 3.

**Synthesis of cross-linked poly(pyrazole–urea) 7c**. To a stirred solution of trifunctional pyrazole **5** (2.507 g, 1 equiv) in anhydrous $CHCl_3$ (2.5 mL) was added HDI (0.634 g, 1.5 equiv). The system was stirred for 2 min, then poured into an aluminum mold (50 mm L × 50 mm W), and allowed to stand at room temperature in a desiccator for *ca.* 24 h. The film was demolded, and placed under vacuum at 70 °C for *ca.* 48 h to ensure complete removal of solvent. The films were kept in a desiccator before measurement.

**Reprocessing experiments**. The cross-linked PPzUs were cut into pieces (~2.8 g) and placed into a rectangular mold (*ca.* 50 mm (L) × 50 mm (W)), and then hot pressed (10 MPa; 140 °C for **7a**, 130 °C for **7b** and **7c**) for 30 min. The mold was cooled to room temperature for ~15 min, and the reprocessed samples were demolded.

## Data availability

The data that support the findings of this study are available from the corresponding authors upon reasonable request.

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

## Acknowledgements

The authors would like to acknowledge the financial support from Ministry of Science and Technology (2017YFB0703300), National Nature Science Foundation of China (51522308, 21572232) and the Chinese Academy of Sciences (QYZDB-SSW-SLH025).

## Author contributions

W.-X. L., N.Z., and J.X. designed the experiments. W.-X. L. performed the experiments and data analysis. W.-X. L., N.Z., and J.X. organized and wrote the paper. Z.Y. and Z.Q. provided valuable comments for the study and paper. DFT calculations were carried out by W.-X. L. with the help of L.Z. and S.L. All authors took part in the discussion of the experiment results.

## Competing interests

The authors declare no competing interests.
