## [Peer Review File · Nature Communications]

Reviewers' comments:

Reviewer #1 (Remarks to the Author):

Report on manuscript NCOMMS-19-18585, titled Dynamic, Multiphase, and Semi-Crystalline Polymers Based on Thermally Reversible Pyrazole-Urea Bonds, by Wen-Xing Liu, Zhusheng Yang, Zhi Qiao, Long Zhang, Ning Zhao, Sanzhong Luo and Jian Xu

In this paper the authors report on new kind of dynamic covalent polyureas, poly(pyrazole-ureas) (PPzUs), which are shown to exhibit excellent mechanical properties and peculiar crystallization behavior, in contrast to reported dynamic polymers, whose crystallization is inhibited because of a different structural design. The first part of the manuscript is devoted to illustrate the reversible character of the pyrazole-urea bonds (PzUBs) through a wide variety of methods, ranging from experimental spectroscopic and thermodynamics technique to theoretical calculation, carried out at DFT level. As detailed by the authors in the second part of the manuscript, the PzUBs reversibility is thereafter exploited to recycle, upon heating, the PPzUs which can be easily prepared, at ambient temperature in the absence of a catalyst, by a highly efficient polyaddition reaction of pyrazoles and diisocyanates.

Given the potential impact of PzUBs in the field of dynamic materials and the growing attention to efficient strategies to improve the structural design of responsive polymers, the topic is in my opinion appropriate for Nature Communications. The manuscript is well written and the results, well supported by both experimental and theoretical evidences, discussed accurately. However, I have few concerns regarding essentially the computational part, and the following remarks should in my opinion be addressed, prior to publication.

- The paragraphs devoted to the discussion of the computational results are presented only at page 10, after the discussion on experimental findings on PPZUs. Since the calculations concern with the PzUBs of the polymer building blocks, whose experimental findings are discussed in the first part of the manuscript, rather than with the polymer itself, it would seem more appropriate to shift the computational results section at the end of such first part, before the section "Synthesis and characterization of poly(pyrazole-ureas)". Otherwise, the computational section seems a bit "appended" at the end of the paper.
- From a computational point of view, my major concern deals with the inclusion of dispersion within the calculations, which is tested by comparing the B3LYP (which does not account for dispersion) and M06-2X (which includes dispersion to a certain extent) functional with the same basis set (6-31+G**). Given the importance, as discussed by the authors, of the aromaticity of the pyrazole unit, do the differences found between the two functionals, and reported in Table S3, can be attributed to such interactions?
- In the main text, it is commented that the differences among the three calculation schemes are not significant. Yet, the free energy for the whole process, computed with the M06-2X functional is more than twice the ones obtained with B3LYP. Conversely, minor variations arise for the free energy of the intermediates. How do the authors explain these features? What is the role of dispersion (which should be the largest difference between B3LHYP and M06-2X) in differentiating the activation (kinetic) free energy from the thermodynamic ones? I think this point deserves some additional comments, either in the main text or in the SI.
- As far as the comparison between the experimental and computed parameters presented in Table S3 is concerned, a qualitative agreement is achieved, but the results are far from being quantitative. Is it due to the role of a different alkylation on the isocyanate (alkyl chain or methyl, for experiment and computation, respectively)? Does neglecting possible local and specific solvent-solute effects (in calculation the solvent DCM is accounted only for through a continuum by PCM) might play some role of the different behavior registered?
- Why where the B3LYP/6-31+G* results shown in the main text and the B3LYP/6-31+G** shifted in the SI? If the authors think the B3LYP is more accurate than M06-2X, the results obtained with the largest basis set should be shown. In any case, they should motivate their choices, otherwise I don't see the point of presenting results with these three particular functionals. Incidentally, the

title within Figure 5 should be corrected, since an asterisk seems to be missing in the basis set specification (6-31+G instead of 6-31+G*). On the contrary, if no polarization function was employed (6-31+G), it should be added in the text.

- In Table S3, it is not clear from the b and d notes, how the free energy is computed for each row. I guess the b and d subscripts in the last 4 rows should be exchanged. Please clarify.

Reviewer #2 (Remarks to the Author):

This manuscript describes synthesis and self-healability of polyurea based on pyrazole-urea bonds. Authors studied dynamic chemistry of pyrazole-urea bonds using small molecule compound and theoretical studies.

The use of pyrazole moieties is not new. In fact, pyrazole has been used in the development of blocked isocyanate materials in coating industries. Also, similar research using hindered urea has been published recently. Furthermore, the temperature required for dynamic chemistry of pyrazole-urea bonds is higher than 110C which may present some limitation of the chemistry to materials applications.

Although the manuscript is well written with experimental data required for self-healability, it would be published in a specific journal.

Reviewer #3 (Remarks to the Author):

The authors prepared dynamic, multiphase, and semi-crystalline polymers based on thermally reversible pyrazole-urea bonds. For the first time, dynamic chemistry of pyrazole-urea bonds was well-studied by using model compounds. The authors also proposed a molecular-level welding concept between dynamic polyurea and polyurethane. The work of this manuscript is practical and logical and should be published after a minor revision. Here are some questions and comments:

1. It is interesting that the uncross-linked PPzU 6 shows a better mechanical strength than cross-linked PPzU 7 (Supplementary Table S2 and Figure 3). This is not usual. Why crosslink? Can PPzU 6 maintain the mechanical strength effectively after the same thermal recycling treatment? Moreover, the yield strength and modulus of PPzU 7a and PPzU 7b had a significant decline after thermal recycling (Supplementary Table S2).
2. The author should not use the "gel fraction \geq 0.97" to demonstrate the densely covalently crosslinking of the materials (Line 118, Page 7). What's more, the PPzU materials in this work can be hardly defined as densely covalently crosslinking, especially for PPzU 7a, whose crosslink density is 233 mol•m⁻³ and swelling ratio is up to 576%.
3. The authors put forward the concept of molecular-level welding. It is not so justified as the experiment was conducted at an elevated temperature and solvent. Welding should be accomplished in solid state. In my view, macromolecular exchange reaction or "polymer scrambling" reaction (J. Am. Chem. Soc. 2003, 125, 4064) is more correct. Also what are the values for the proposed molecular-level welding?
4. There is an error in Line 135, Page 8, the "PPzU 5" should be "PPzU 6".
5. Supplementary Figure 10 (D) was not clear and the soaking time in solvent was not given. It is better if PPzU 8, which has a lower crystalline degree, can be tested as a contrast sample.
6. In the introduction part, about the review on dynamic urea bond: Except for introducing bulky substituents to make urea bond dynamic (Nat. Commun. 2014, 5, 3218), the addition of metal ion is also effective to reduce the dissociation activation energy of urea bond (J. Mater. Chem. A, 2019, 7, 15933-15943).

Dear reviewers,

On behalf of my co-authors, we thank all the reviewers for the helpful comments, which have helped us to greatly improve our manuscript. We carefully considered all reviewers' comments and did our best to address these concerns. In the text below we provide a point-by-point response to all comments made by the three reviewers. The changes were marked in the revised Manuscript and Supplementary Information.

Thank you and best regards.

Reviewer #1 (Remarks to the Author):

Report on manuscript NCOMMS-19-18585, titled Dynamic, Multiphase, and Semi-Crystalline Polymers Based on Thermally Reversible Pyrazole-Urea Bonds, by Wen-Xing Liu, Zhusheng Yang, Zhi Qiao, Long Zhang, Ning Zhao, Sanzhong Luo and Jian Xu

In this paper the authors report on new kind of dynamic covalent polyureas, poly(pyrazole-ureas) (PPzUs), which are shown to exhibit excellent mechanical properties and peculiar crystallization behavior, in contrast to reported dynamic polymers, whose crystallization is inhibited because of a different structural design. The first part of the manuscript is devoted to illustrate the reversible character of the pyrazole-urea bonds (PzUBs) through a wide variety of methods, ranging from experimental spectroscopic and thermodynamics technique to theoretical calculation, carried out at DFT level. As detailed by the authors in the second part of the manuscript, the PzUBs reversibility is thereafter exploited to recycle, upon heating, the PPzUs which can be easily prepared, at ambient temperature in the absence of a catalyst, by a highly efficient polyaddition reaction of pyrazoles and diisocyanates.

Given the potential impact of PzUBs in the field of dynamic materials and the growing attention to efficient strategies to improve the structural design of responsive polymers, the topic is in my opinion appropriate for Nature Communications. The manuscript is well written and the results, well supported by both experimental and theoretical evidences, discussed accurately. However, I

have few concerns regarding essentially the computational part, and the following remarks should in my opinion be addressed, prior to publication.

Response: We appreciate the reviewer for the time reviewing our manuscript and the positive comments. The manuscript has been carefully revised according to the suggestions from the reviewer.

- The paragraphs devoted to the discussion of the computational results are presented only at page 10, after the discussion on experimental findings on PPZUs. Since the calculations concern with the PzUBs of the polymer building blocks, whose experimental findings are discussed in the first part of the manuscript, rather than with the polymer itself, it would seem more appropriate to shift the computational results section at the end of such first part, before the section “Synthesis and characterization of poly(pyrazole-ureas)”. Otherwise, the computational section seems a bit “appended” at the end of the paper.

Response: Many thanks for this suggestion. We have made this shift.

- From a computational point of view, my major concern deals with the inclusion of dispersion within the calculations, which is tested by comparing the B3LYP (which does not account for dispersion) and M06-2X (which includes dispersion to a certain extent) functional with the same basis set (6-31+G**). Given the importance, as discussed by the authors, of the aromaticity of the pyrazole unit, do the differences found between the two functionals, and reported in Table S3, can be attributed to such interactions?

Response: Thanks. We do think the aromaticity of the pyrazole plays an important role in the easy dissociation of the C-N bond. However, it looks like that the difference between the two functionals is mainly caused by the C-N bond formation process. The following table lists the free energies of these intermediates and transition states of the two functionals (Table R1). As we can see, the differences between the free energies of the two functionals (ΔG_1) significantly decrease during and after the C-N bond formation, indicating that the computed bond energy difference

should be the main contribution to the divergence between the two functionals.

Table R1. The free energy differences of the reaction species between the two computational methods.

Species	G (B3LYP, Hartree) ^a	G (M06-2X, Hartree) ^a	ΔG_1 (kcal·mol ⁻¹) ^b
MeNCO	-207.9837495	-207.8875714	60.35272
pyrazole	-226.1872519	-226.0866745	63.11324
Int1	-434.1609842	-433.9635922	123.8653
TS1	-434.1333699	-433.9419676	120.1068
Int2	-434.1418345	-433.9531814	118.3816
TS2	-434.1375545	-433.947277	119.401
P1	-434.1791091	-433.9931388	116.6982

^a Calculated with 6-31+G(d,p) basis set. ^b $\Delta G_1 = G$ (M06-2X) – G (B3LYP).

We also investigated the influence of the aromaticity and internal hydrogen bond on the stability of the final product by comparing the energy of the following three structures (Table R2). In **R1**, the influence of the aromaticity effect on carbonyl moiety is ruled out while the internal hydrogen bond is retained. In **R2**, the internal hydrogen bond is removed while the aromaticity effect is retained. In **R3**, both the aromaticity effect and internal hydrogen bond are removed. The enthalpy and free energy of **R1** are 0.77 and 1.07 kcal·mol⁻¹ lower than **P1**, respectively, indicating the aromaticity of the pyrazole indeed diminishes the conjugation effect between the pyrazole nitrogen and the carbonyl group. The energy of **R2** is much larger than **P1**. On the one hand, the removal of the internal hydrogen bond can increase the energy; on the other hand, the repulsion between the lone electron pairs on Nitrogen and Oxygen also causes the destabilization. As we can see in **R3**, no lone electron pair repulsion exists, and the energy of **R3** is comparable with **P1**. These results indicate that the aromaticity and internal hydrogen bonding only show little influence on the stability of the final product. Therefore, we don't think aromaticity is the main contribution to the divergence between the two functionals.

Table R2. Enthalpy and free energy of **P1** and **R1–3**.

					P1	R1	R2	R3
			H (M06-2X) ^a	G (M06-2X) ^a
	P1 (Hartree)		-433.9493738	-433.9931388
	R1 (Hartree)		-433.950601	-433.994839

R2 (Hartree)	-433.941853	-433.982885
R3 (Hartree)	-433.948908	-433.992107
R1-P1 (kcal·mol ⁻¹)	-0.7701	-1.0669
R2-P1 (kcal·mol ⁻¹)	4.71936	6.43434
R3-P1 (kcal·mol ⁻¹)	0.29228	0.64745

^a Calculated with 6-31+G(d,p) basis set.

• In the main text, it is commented that the differences among the three calculation schemes are not significant. Yet, the free energy for the whole process, computed with the M06-2X functional is more than twice the ones obtained with B3LYP. Conversely, minor variations arise for the free energy of the intermediates. How do the authors explain these features? What is the role of dispersion (which should be the largest difference between B3LYP and M06-2X) in differentiating the activation (kinetic) free energy from the thermodynamic ones? I think this point deserves some additional comments, either in the main text or in the SI.

Response: As we stated above, although the main difference between M06-2X and B3LYP is the treatment of dispersion, we don't think it is the reason for the large free energy difference with M06-2X. The involvement of dispersion is indeed very important when dealing with weak interaction, but in this reaction system, no significant weak interaction exists. Table R1 shows that the free energy differences between M06-2X and B3LYP (ΔG_1) significantly decrease during and after the C-N bond formation (**P1** << **TS2**, **Int2** < **TS1** << **Int1** \approx MeNCO+pyrazole). The similar free energy differences (ΔG_1) for hydrogen-bonded **Int1** and starting materials (MeNCO+pyrazole) may exclude the possibility that dispersion has a big influence on this system. Instead, the strong C-N bond formation accounts for the large difference of the thermodynamic free energy (ΔG_b). We have added the comments in the Supplementary Information (Supplementary Table 2).

• As far as the comparison between the experimental and computed parameters presented in Table S3 is concerned, a qualitative agreement is achieved, but the results are far from being quantitative. Is it due to the role of a different alkylation on the isocyanate (alkyl chain or methyl, for experiment and computation, respectively)? Does neglecting possible local and specific solvent-solute effects (in calculation the solvent DCM is accounted only for through a continuum

by PCM) might play some role of the different behavior registered?

Response: Thanks for the suggestion. We think the different alkyl substituents may cause a certain difference between experiment and computation results. In addition, the small basis set used in the manuscript can also cause some inaccuracy. We further calculated the energy using a bigger basis set (6-311++G(2df,2pd)) with SMD solvation model (Fig. 2a and Supplementary Fig. 10). The activation energy of the reversed reaction significantly decreases (Supplementary Table 1). We apologize for missing a statement in the Computational Methods, that the solvation model we used is CPCM with SMD-coulomb atomic radii (page 4 in the Supplementary Information). We have also examined the SMD model with B3LYP and M06-2X methods, and no significant difference was found.

- Why were the B3LYP/6-31+G* results shown in the main text and the B3LYP/6-31+G** shifted in the SI? If the authors think the B3LYP is more accurate than M06-2X, the results obtained with the largest basis set should be shown. In any case, they should motivate their choices, otherwise I don't see the point of presenting results with these three particular functionals. Incidentally, the title within Figure 5 should be corrected, since an asterisk seems to be missing in the basis set specification (6-31+G instead of 6-31+G*). On the contrary, if no polarization function was employed (6-31+G), it should be added in the text.

Response: We are sorry for the mistake. The results shown in the main text were B3LYP/6-31+G(d), and now we have updated the data to the level of B3LYP/6-311++G(2df,2pd)//B3LYP/6-31+G(d,p) (Fig. 2a). As mentioned in the Computational Methods (page 4 in the Supplementary Information), we employed B3LYP/6-31+G(d) for all calculations including the favored pathway, disfavored pathway, and resonance energy analysis. Besides, for the favored pathway we also used M06-2X functional for comparison and bigger basis sets to improve accuracy (Fig. 2a and Supplementary Fig. 10).

- In Table S3, it is not clear from the b and d notes, how the free energy is computed for each row. I guess the b and d subscripts in the last 4 rows should be exchanged. Please clarify.

Response: The subscript of b refers to the binding reaction (forward reaction) and d refers to the dissociation reaction (backward process). The activation free energies ΔG_b^\ddagger and ΔG_d^\ddagger are calculated from Supplementary Figs 3 and 9, respectively. The reaction free energy ΔG_b is calculated from Supplementary Fig 5. We have clarified these in Supplementary Table 1 in the revised Supplementary Information

Reviewer #2 (Remarks to the Author):

This manuscript describes synthesis and self-healability of polyurea based on pyrazole-urea bonds. Authors studied dynamic chemistry of pyrazole-urea bonds using small molecule compound and theoretical studies.

The use of pyrazole moieties is not new. In fact, pyrazole has been used in the development of blocked isocyanate materials in coating industries. Also, similar research using hindered urea has been published recently. Furthermore, the temperature required for dynamic chemistry of pyrazole-urea bonds is higher than 110C which may present some limitation of the chemistry to materials applications.

Response: We thank the reviewer very much for spending precious time on our manuscript. We would like to further explain the novelty and importance of the manuscript by the following responses to the comments of the reviewer.

It is of great scientific and technological importance to discover the dynamic nature of old chemistry and then utilize it for dynamic polymer materials. Although pyrazoles have been studied as blocked molecules for isocyanate, the pyrazole-urea bonds have not been applied for the design of dynamic polymers with reprocessing/recyclable capacities. What's more, the kinetics, thermodynamics and reversible mechanism of pyrazole-urea bonds were not intensively investigated before. The dynamic nature of pyrazole-urea bonds is a significant contribution to the fields of dynamic polymer materials. The detailed reaction parameters of pyrazole-urea bonds from both experimental and theoretical aspects may also be valuable to other authors that use

dynamic covalent bonds, such as dynamic combinatorial chemistry, self-assembly, shape-memory materials, 3D printing, *etc.*

Cheng *et al.* reported interesting hindered urea bonds which are significantly destabilized by bulky substituents and reversible at room temperature. The hindered urea bonds are used for the design of self-healing/recyclable amorphous polyureas, which are mechanically weak organogels (*Nat. Commun.* **2014**, *5*, 3218) or hard plastics with low extensibility (*Adv. Mater.* **2016**, *28*, 7646-7651). These polymers based on hindered urea bonds suffer from hydrolysis of the isocyanate intermediate at ambient temperature. In contrast to hindered urea bonds, the pyrazole moiety is introduced into urea bonds in our work. Owing to the characteristic aromatic nature of pyrazole, pyrazole-ureas are stable at room temperature and reversible upon heating. The dynamic mechanism is attributed to the opposed resonance effect of pyrazole and the N-assisting intramolecular hydrogen transfer. Moreover, as the rational design of multiphase structure is employed for the polymers, poly(pyrazole-ureas) have unique crystallization ability, which gives excellent strength, high extensibility, and solvent resistance. Additionally, the concept of molecular-level blending is introduced and two different species of polymers, polyurethane and polyurea, are blended for the first time by this method. Therefore, our work on pyrazole-urea bonds is different from Cheng's published hindered urea bonds, in terms of dynamic structure, reversible mechanism, stability, polymer design, polymer properties, and the applications of dynamic covalent bonds.

Besides the dynamic chemistry, like hindered urea bonds, happening at room temperature, we would like to emphasize that thermally reversible covalent bonds are also critically important for dynamic materials. To name a few, Diels–Alder reaction of furans with maleimides (*Science* **2002**, *295*, 1698-1702), transesterification (*Science* **2011**, *334*, 965-968), vinylogous urethane (*Nat. Commun.* **2017**, *8*, 14857) are typically reversible at 120 °C, 180 °C, and 100 °C, respectively. They have better thermal resistance and can withstand higher temperatures without loss of strength or change of structure, which allows for the applications requiring a higher temperature.

Although the manuscript is well written with experimental data required for self-healability, it would be published in a specific journal.

Response: We appreciate that the referee recognizes the overall quality of the manuscript. As described above, our work is of interest for chemists and materials scientists. The quality of the manuscript has been improved after addressing the comments from the referees. We believe that the revised manuscript accomplishes the requisite of generality and broad interest for the community of Nature Communications.

Reviewer #3 (Remarks to the Author):

The authors prepared dynamic, multiphase, and semi-crystalline polymers based on thermally reversible pyrazole-urea bonds. For the first time, dynamic chemistry of pyrazole-urea bonds was well-studied by using model compounds. The authors also proposed a molecular-level welding concept between dynamic polyurea and polyurethane. The work of this manuscript is practical and logical and should be published after a minor revision. Here are some questions and comments:

Response: We thank the reviewer for the positive comments on our work. We are glad that the reviewer finds this work important and practical.

1. It is interesting that the uncross-linked PPzU 6 shows a better mechanical strength than cross-linked PPzU 7 (Supplementary Table S2 and Figure 3). This is not usual. Why crosslink? Can PPzU 6 maintain the mechanical strength effectively after the same thermal recycling treatment? Moreover, the yield strength and modulus of PPzU 7a and PPzU 7b had a significant decline after thermal recycling (Supplementary Table S2).

Response: We agree with this reviewer that usually crosslinking could enhance mechanical strength. In our polymers, besides the chemical crosslinking, crystallization which serves as physical crosslinking also has a significant contribution to the mechanical properties of PPzUs. For highly covalently crosslinked PPzUs 7, crystallization is strongly inhibited, which leads to a decrease in tensile strength. The similar phenomenon that covalently crosslinked polymers shows lower mechanical strength than the linear ones can also be found in crystalline polyethylene (*Polymer* **1982**, 23, 1944-1952). Compared with thermoplastic polymers, covalently crosslinked

thermosets usually have excellent solvent resistance and thermal stability. In our work, PPzUs **7** are not soluble in solvents including CH_2Cl_2 and CHCl_3 , in which PPzU **6** dissolve. The TGA and DMA experiments also prove that the thermostability of PPzUs **7** is better than that of linear **6** (Supplementary Figs 16 and 19).

Figure R1. Stress-strain curves for the original and the recycled PPzU **6** (10 MPa, 130 °C, 30 min).

Actually, the as-synthesized original PPzU **6** has been thermally treated (10 MPa, 130 °C, 30 min) to produce a defect-free film from the granules (see the Supplementary Methods). We cut PPzU **6** film into pieces and performed thermal recycling treatment again (Figure R1). The mechanical property of the recycled PPzU **6** (Young's Modulus = 206.8 ± 1.1 MPa, Yield Stress = 29.5 ± 0.2 MPa, Stress of break = 35.3 ± 0.6 MPa, Strain of break = $828 \pm 17\%$) is comparable to that of the original sample. The result shows that the mechanical properties are effectively maintained.

As discussed in the manuscript, the significant decline on yield stress and modulus of the crosslinked PPzUs after the first thermal recycling can be attributed to the distinction of crystallization behavior between the original samples and the recycled samples. In other words, the original samples with higher crystallinity show higher yield stress and modulus than the recycled samples with lower crystallinity. This difference in crystallization behavior has been proved by the SAXS and XRD experiments (Supplementary Figs 20 and 26).

2. The author should not use the “gel fraction \geq 0.97” to demonstrate the densely covalently

crosslinking of the materials (Line 118, Page 7). What's more, the PPzU materials in this work can be hardly defined as densely covalently crosslinking, especially for PPzU 7a, whose crosslink density is $233 \text{ mol}\cdot\text{m}^{-3}$ and swelling ratio is up to 576%.

Response: We thank the reviewer for this valuable suggestion. We have clarified this in the revised version of our paper with the following sentence:

“Of particular note is that PPzU 7c with the highest crosslinking degree (average molecular weight between crosslinks (M_c) = $1.7 \text{ kg}\cdot\text{mol}^{-1}$, Supplementary Table 3) is still semi-crystalline ...”

3. The authors put forward the concept of molecular-level welding. It is not so justified as the experiment was conducted at an elevated temperature and solvent. Welding should be accomplished in solid state. In my view, macromolecular exchange reaction or “polymer scrambling” reaction (J. Am. Chem. Soc. 2003, 125, 4064) is more correct. Also what are the values for the proposed molecular-level welding?

Response: Many thanks for this insightful suggestion. More widely used term, “blending”, is now used instead of “welding” throughout the manuscript. We think this molecular-level blending is a new technique to make polymer blends or hybrid materials. As we know, different species of polymers are immiscible in most cases. Owing to the introduction of reversible covalent bonds, polymer chains from different species of polymers can exchange. Therefore, this technique can be applied to synthesize polymer mixture with better compatibility. For example, recently enhanced adhesion of poly(methyl methacrylate) and polyethylene was achieved through dynamic dioxaborolane metathesis reaction (*Science* **2017**, 356, 62-65). In addition, the macromolecular exchange reaction can control topological transformations of polymers (*Nat. Chem.* **2017**, 9, 817-823). We have modified the corresponding statement to make it clear in our revised manuscript as follows:

“Different from traditional polymer blending methods, this artful methodology using dynamic covalent bonds provides a new approach to polymer blends.”

4. There is an error in Line 135, Page 8, the “PPzU 5” should be “PPzU 6”.

Response: Thanks. The mentioned error has been corrected in our revised manuscript.

5. Supplementary Figure 10 (D) was not clear and the soaking time in solvent was not given. It is better if PPzU 8, which has a lower crystalline degree, can be tested as a contrast sample.

Response: As suggested, Supplementary Fig. 15D was updated to make it clear and the soaking time was also given. Solubility test of PPzU 8 is showed in Supplementary Fig. 15E. As expected, the solvent resistance of PPzU 6 is much better than that of PPzU 8.

6. In the introduction part, about the review on dynamic urea bond: Except for introducing bulky substituents to make urea bond dynamic (Nat. Commun. 2014, 5, 3218), the addition of metal ion is also effective to reduce the dissociation activation energy of urea bond (J. Mater. Chem. A, 2019, 7, 15933-15943).

Response: We thank the reviewer for suggesting this relevant reference. In the introduction part, we have introduced this work (page 3) as follows:

“Metal catalyst such as zinc acetate shows effectiveness to increase the reversibility of normal urea bonds.⁴⁰”

REVIEWERS' COMMENTS:

Reviewer #1 (Remarks to the Author):

Report on the revised version of manuscript NCOMMS-19-18585, titled "Dynamic, Multiphase, and Semi-Crystalline Polymers Based on Thermally Reversible Pyrazole-Urea Bonds",
by Wen-Xing Liu, Zhusheng Yang, Zhi Qiao, Long Zhang, Ning Zhao, Sanzhong Luo and Jian Xu

The manuscript is well written and the results are accurately discussed and the conclusions supported by both experimental and computational evidences.
IN the revised version, the authors have answered to all of my previous remarks, discussing my criticisms and changing the text accordingly. For these reasons, the paper is in my opinion suitable for publication in NatureCommunications in its present form.

Reviewer #2 (Remarks to the Author):

I believe that authors have addressed the comments from the reviewers. I recommend to be published as it is.

Reviewer #3 (Remarks to the Author):

The authors has done a very good revision according to the reviewers' questions and suggestions. In my view, now it is acceptable in the reputed journal "Nature Communications".

Reviewer #1 (Remarks to the Author):

Report on the revised version of manuscript NCOMMS-19-18585, titled

"Dynamic, Multiphase, and Semi-Crystalline Polymers Based on Thermally Reversible Pyrazole-Urea Bonds",

by Wen-Xing Liu, Zhusheng Yang, Zhi Qiao, Long Zhang, Ning Zhao, Sanzhong Luo and Jian Xu

The manuscript is well written and the results are accurately discussed and the conclusions supported by both experimental and computational evidences.

IN the revised version, the authors have answered to all of my previous remarks, discussing my criticisms and changing the text accordingly. For these reasons, the paper is in my opinion suitable for publication in Nature Communications in its present form.

Reviewer #2 (Remarks to the Author):

I believe that authors have addressed the comments from the reviewers. I recommend to be published as it is.

Reviewer #3 (Remarks to the Author):

The authors has done a very good revision according to the reviewers' questions and suggestions.

In my view, now it is acceptable in the reputed journal "Nature Communications".

Response: We thank the Reviewers very much for recommending publication of our work and greatly appreciate their comments during the reviewing process.